# TADABENCH-1M: A LARGE-SCALE WET-LAB PROTEIN BENCHMARK FOR RIGOROUS OOD EVALUATION

## ABSTRACT

Existing benchmarks for biological language models (BLMs) inadequately capture the challenges of real-world applications, often lacking realistic out-of-distribution (OOD) scenarios, evolutionary depth, and consistency in measurement. To address this, we introduce TadABench-1M, a new benchmark based on a wet-lab dataset of over one million variants of the therapeutically relevant TadA enzyme, purpose-built to embody these three essential attributes. Generated across 31 rounds of wet-lab evolution, it offers unparalleled evolutionary depth and naturally presents a stringent OOD challenge. To ensure measurement consistency across this extensive campaign, we developed Seq2Graph, a scalable graph-based algorithm that systematically unifies multi-batch experimental data. Our high-fidelity benchmark highlights a critical finding: while state-of-the-art BLMs excel on a standard random split of the data (Spearman's $\rho \approx 0.8$), they fail dramatically on a realistic temporal prediction task ($\rho \approx 0.1$). This stark performance gap validates the importance of our benchmark's design principles and suggests that evolutionary depth is critical for building models with realistic utility.

## 1 INTRODUCTION

Language models pre-trained on biomolecular sequences—DNA, RNA, and proteins—have recently exhibited scaling laws (Madani et al., 2023; Chen et al., 2024; Ji et al., 2021). As parameter count and corpus size grow, the computational evaluation, such as perplexity and masked-token accuracy, steadily improves. However, despite this in-silico success, their practical utility faces increasing scrutiny. A recent crowdsourced antibody-design challenge, for example, found that models with state-of-the-art computational metrics failed to generate high-affinity binders (Cotet et al., 2025). This highlights a critical disconnect between standard benchmarks and real-world outcomes.

Addressing this gap requires a new class of benchmarks designed to mirror the challenges of real-world protein engineering. We argue that such a benchmark must embody three key attributes as shown in Figure 1:

1. **Out-of-Distribution Challenge**: Protein engineering inherently seeks novel variants, demanding that models generalize to unseen regions of the sequence space.

2. **Evolutionary Depth**: Unlike one-shot mutation scanning, engineering is an iterative process. A benchmark must capture this trajectory, requiring models to extrapolate across sequences with accumulating mutations—a challenge we term evolutionary depth.

3. **Measurement Consistency**: Large-scale, multi-round experiments are prone to batch effects and noise. A reliable benchmark must therefore ensure that fitness labels are consistent and comparable across the entire evolutionary campaign.

While existing benchmarks like ProteinGYM (Notin et al., 2024) and ProteinBench (Ye et al., 2024) have been invaluable, they typically prioritize breadth (sampling diverse protein families) over the mutational depth needed to simulate a focused, iterative engineering project. As a result, a model's utility in guiding a multi-step design cycle, a critical real-world capability, remains largely untested.

To address these challenges, we introduce TadABench-1M, a new benchmark built upon a wet-lab dataset of over one million variants of tRNA-specific adenosine deaminase (TadA), the catalytic

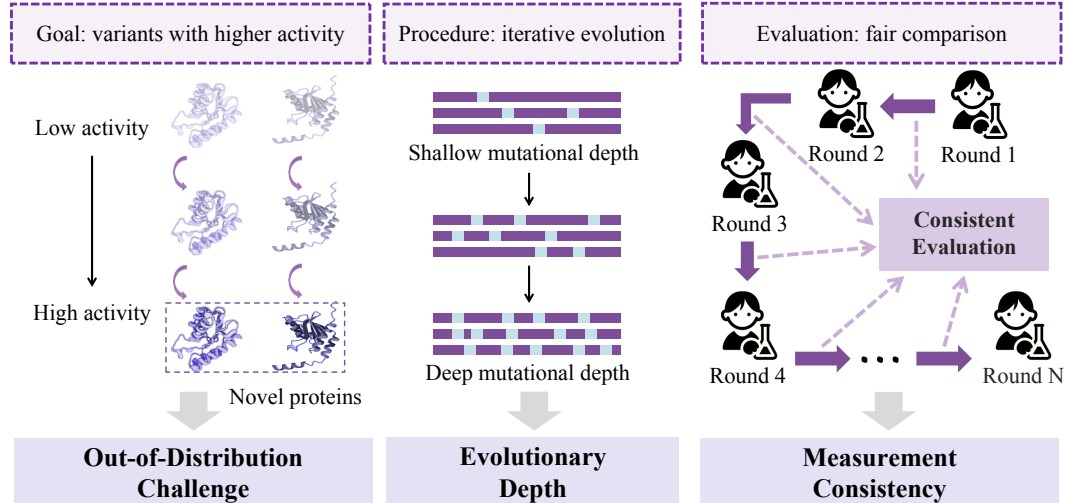

Figure 1: Engineering for TadA protein is an iterative process of creating new protein variants with higher activity. This process demands three key capabilities: realistic out-of-distribution generalization, extrapolation across evolutionary depth, and high measurement consistency. We introduce TadABench-1M as a benchmark built upon these attributes.

engine of adenine base editors with significant therapeutic value (Komor et al., 2016; Gaudelli et al., 2017). It is specifically designed to embody the three key attributes we identified. By conducting 31 iterative rounds of directed evolution, the dataset provides immense evolutionary depth with variants containing up to 25 mutations. This multi-round process also naturally establishes a stringent OOD challenge: predicting fitness in future rounds using data from the past. To ensure measurement consistency across such an extensive campaign, we develop Seq2Graph, a method that unifies fitness labels by robustly correcting for inter-round batch effects and mitigating experimental noise, a critical step for ensuring the reliability of this large-scale benchmark.

The benchmark is designed around a practical temporal split that mirrors a real-world protein engineering campaign, where the task is to predict the activity of variants in a future round of evolution given data from all prior rounds. We demonstrate that while state-of-the-art BLMs perform well on a conventional i.i.d. random split of our data ($\rho \approx 0.8$), their performance collapses on our temporal split, yielding a Spearman correlation of only $\rho \approx 0.1$. Furthermore, our analysis reveals a critical insight into data curation: generalization performance is primarily driven by the evolutionary depth of the training data (*i.e.*, the number of evolutionary rounds covered), not its raw volume. Finally, we demonstrate the benchmark's data consistency by both in vitro and in silico experiments.

**Contributions.**

- We introduce TadABench-1M, a new benchmark built upon a million-scale protein activity dataset from 31 rounds of wet-lab evolution. To ensure its measurement consistency, we present Seq2Graph, a scalable method for unifying and de-noising fitness labels across multi-round experiments.

- By incorporating a temporal split of the dataset, TadABench-1M establishes a realistic evaluation protocol that mimics a real-world engineering campaign and reveals a stark generalization gap: SOTA BLMs fail ($\rho \approx 0.1$) despite excelling on a standard random split ($\rho \approx 0.8$).

- Through systematic analysis, we demonstrate that sequence diversity and evolutionary depth are more critical for OOD generalization than raw data volume. These results offer clear guidance for future experimental design and data curation in real-world applications such as protein engineering.

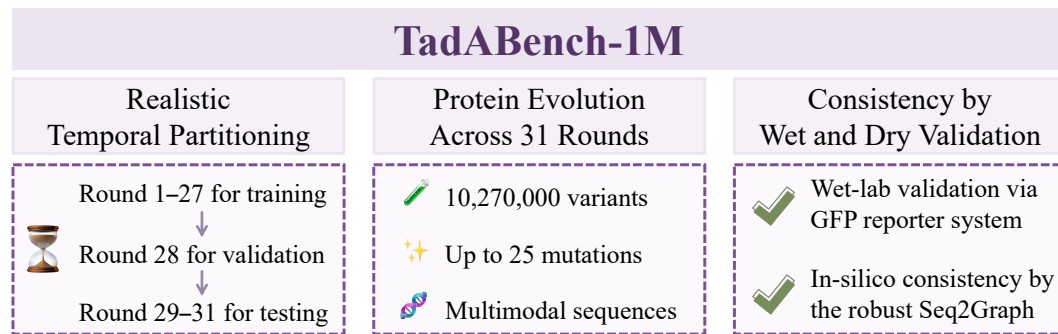

Figure 2: TadABench-1M simulates the realistic engineering process. It adopts a temporal data split to reflect practical demands. Across 31 rounds of evolution, it includes over one million variants with up to 25 mutations. Activity consistency is ensured through wet-lab experiments and Seq2Graph.

## 2 RELATED WORK

### 2.1 PROTEIN ACTIVITY BENCHMARK

While structural benchmarks in protein research have achieved notable success (Ye et al., 2024; Moult et al., 2020; Haas et al., 2018; Buttenschoen et al., 2024), functional benchmarks are still in nascent stages. These benchmarks are primarily categorized into two groups (West-Roberts et al., 2024), biophysical properties and deep mutational scanning (DMS) data. The benchmarks for biological properties (Bairoch, 2000; Xu et al., 2022; Zhou et al., 2019; Nikam et al., 2021; Rao et al., 2019; Vander Meersche et al., 2024) include metrics like enzymatic activity, fluorescence, thermodynamics, and solubility; however, their broad focus limits their utility in precise evaluations. DMS benchmarks (Fowler & Fields, 2014; Jiang et al., 2024; Gray et al., 2018), which utilize large-scale mutagenesis and high-throughput sequencing, offer detailed insights into fitness landscapes for protein mutations. Researchers (Notin et al., 2024; Dallago et al., 2021; Riesselman et al., 2018) also leverage diverse DMS datasets to construct the comprehensive benchmark, but may introduce inconsistency since the way data is treated varies widely within the community (Notin et al., 2024). As shown in Figure 2, TadABench-1M overcomes these limitations by uniquely combining three key features: a realistic OOD challenge, high sequence diversity reflecting evolutionary depth, and strict measurement consistency guaranteed by standardized wet-lab experiments and our Seq2Graph.

### 2.2 BASE EDITING DATASET

Current datasets focusing on deaminase enzyme optimization in base editing are relatively sparse and often fragmented. Most available resources (Dixit et al., 2024; Yan et al., 2020; Marquart et al., 2021; Sánchez-Rivera et al., 2022; Xiang et al., 2021; Leenay et al., 2019) emphasize the optimization of base editors through the lens of their interactions with CRISPR-associated proteins (Cas) and single-guide RNAs (sgRNAs), rather than through a systematic exploration of the deaminase variants themselves. These datasets are typically derived from narrow experimental conditions, thereby limiting their generalizability and scalability for machine learning (ML)-based modeling and prediction. Several research groups have published improved or novel deaminase protein sequences through directed evolution or rational design (Richter et al., 2020; Li et al., 2020; Tu et al., 2022; Perrotta et al., 2024; Cheng et al., 2024), but with various wet-lab experimental conditions. Recent aggregation platforms such as CRISPRbase (Fan et al., 2023) aim to centralize base editing datasets across various publications and labs. While this is a significant step forward in data accessibility, it introduces significant batch effects due to diverse experimental protocols (Notin et al., 2024), compromising the accuracy of models under real experimental conditions.[1]

---

[1]We have used large language models to polish writing in this paper.

## 3 DATASET CONSTRUCTION BY SEQ2GRAPH

Constructing a million-scale, well-annotated fitness landscape presents three primary challenges: (1) generating a large number of meaningful fitness labels within each experimental round, (2) acquiring data from a sufficient number of distinct evolution rounds to ensure dataset diversity and depth, and (3) developing a robust algorithm to unify labels across these independent experiments.

While the first two challenges are met through our wet-lab protocol (detailed in Appendix B), this paper focuses on the third, a computational challenge: developing a robust algorithm to unify labels across these independent experiments. This section details our cross-experiment dataset construction method, Seq2Graph, illustrated in Figure 3. We begin by presenting the motivation for our approach (Section 3.1). We then describe the three key stages of our method: directed graph construction (Section 3.2), inconsistency elimination (Section 3.3), and activity assignment (Section 3.4).

### 3.1 MOTIVATION

Conventional fitness quantification methods have proven effective for single-batch experiments, typically relying on normalized read counts to measure variant activity (Wagner et al., 2012; Love et al., 2014; Robinson & Oshlack, 2010). However, they face significant limitations in deep evolutionary experiments that involve multiple rounds, especially when there is only partial overlap of the variant sets between rounds. The inherent stochasticity and batch effects in protein engineering experiments introduce statistical bias when normalizing across rounds. Specifically, variants observed in fewer rounds are based on fewer data points, which leads to inaccurate estimations of their activity scores.

To overcome this challenge, we propose focusing on **relative activity rankings within each round**. This strategy avoids the biases associated with direct normalization, offering a more reliable signal for integrating data across rounds. Given the scale and depth of our dataset—comprising 31 evolutionary rounds and over a million sequences—graph-based algorithms emerge as an ideal approach for constructing a globally consistent fitness landscape. Our framework, Seq2Graph, leverages this methodology by linking overlapping variants, forming a connected graph of relative activities. This methodology offers a general solution for benchmarking in experiments with multiple rounds of data.

### 3.2 DIRECTED GRAPH CONSTRUCTION

In our wet-lab experiments, protein variants with enhanced activity result in increased read count enrichment during next-generation sequencing (NGS)[2] of the final populations, shown on the left of Figure 3. We model the relative activity of variants as a directed graph $G = (V, E)$, where each unique DNA sequence represents a node $v \in V$.

For each of the 31 rounds, we first rank all variants by their read count enrichment. To construct the edges, we create a directed edge $e_{i \to j}$ only between variants $v_i$ and $v_j$ if they are adjacent in this sorted list. This design choice is critical for two reasons: (1) Reliability: adjacent variants have similar read counts, making their ratio a more reliable and less noisy local signal compared to the ratio between distant ranks. (2) Sparsity: this approach generates a sparse graph, ensuring the computational tractability of subsequent steps.

Each edge points from the variant with the higher growth multiple ($v_i$) to the one with the lower multiple ($v_j$), and its weight $w_{ij}$ is defined as $w_{ij} = C(v_i)/C(v_j) > 1$, where $C(\cdot)$ is the read count enrichment after each independent evolution. The final graph, aggregating edges from all 31 rounds, is a single connected component due to sequence overlap between rounds.

### 3.3 INCONSISTENCY ELIMINATION

Inconsistencies arising from experimental noise across the 31 rounds manifest as cycles in the aggregated graph in Figure 3 (b). For global consistency of activities, we resolve conflicts by transforming the graph into a Directed Acyclic Graph (DAG). This is achieved by removing a

---

[2]Next-Generation Sequencing (NGS) is a broad technology platform that enables deep sequencing, which refers to generating high coverage of sequencing reads for a target region. In this paper, we use NGS and deep sequencing interchangeably.

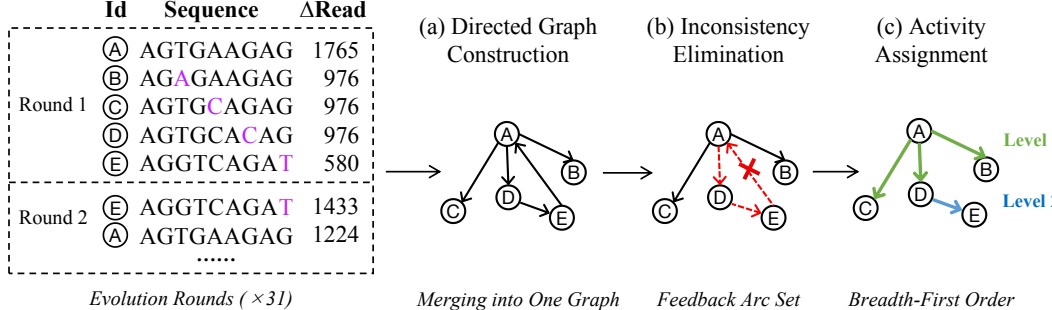

Figure 3: Pipeline of Seq2Graph. (a) We integrate deep sequencing data from 31 independent rounds of evolution into a directed acyclic graph. (b) We apply the Feedback Arc Set algorithm to resolve inconsistencies across experimental rounds. (c) We assign activity values to each TadA variant following a breadth-first traversal order to minimize sequencing noise along relative relationships.

minimum-weight set of edges to break all cycles. We formalize this as the weighted Feedback Arc Set (FAS) problem, where we aim to remove edges with the smallest weights:

$$
\min_{F \subseteq E} \quad \sum_{e \in F} w_e
$$

$$
\text{s.t.} \quad G' = (V, E \setminus F) \text{ is acyclic}
$$

(1)

Since the FAS problem is NP-hard, we employ a greedy heuristic (Eades et al., 1993) for fast approximation. For computational efficiency, we first identify all strongly connected components (SCCs) in the graph. Since cycles can only exist within SCCs, we apply the heuristic independently to each SCC subgraph, which is significantly faster than processing the entire graph.

## 3.4 ACTIVITY ASSIGNMENT

With a DAG established, we assign a final activity score to each node in Figure 3(c). First, we anchor the landscape by assigning a reference activity of 1.0 to the sequence TadA8e (Richter et al., 2020). We then propagate activity values throughout the graph. Since edge weights represent activity ratios and biological growth is multiplicative, this propagation is performed in the log domain.

We traverse the graph from the reference node. For any node reachable via multiple paths, its activity is determined by the shortest path. This strategy prioritizes the most direct comparisons and mitigates the accumulation of noise that can occur over longer paths. We implement this assignment by using a Breadth-First Search (BFS) and disregarding edge directions. This process yields the final DNA-level dataset. To derive protein-level activities, we average the scores of all synonymous DNA variants encoding the same protein sequence. This aggregation is non-trivial, as these variants often exhibit subtle activity differences due to effects like codon usage bias.

## 3.5 DISCUSSION OF METHODOLOGY

**Rationale for Guided Exploration Data.** A key feature of our dataset is its origin from a guided, multi-round TadA engineering rather than uniform random mutagenesis. In practice, it is infeasible to perform saturation mutagenesis across 25+ positions ($> 10^{15}$ variants). Effective engineering leverages expert knowledge to explore promising regions of the sequence space. This knowledge-driven process is precisely the real-world scenario we aim to benchmark for BLMs.

**Generality of the Seq2Graph Framework.** Seq2Graph is a task-agnostic and generalizable framework. Its core principle of reconciling locally-consistent rankings via a graph is not specific to our protein system. It is applicable to any high-throughput screen that generates quantitative readouts across multiple independent experiments, batches, or even different labs. For example, we have also built a Cas9 protein dataset using the same algorithm, as shown in Section C.4.

# 4 EXPERIMENT

In this section, we present a series of experiments designed to validate TadABench-1M as a challenging and reliable benchmark for guiding real-world protein directed evolution campaigns. **Out-of-distribution:** We first establish that our proposed temporal data split poses a significant out-of-distribution (OOD) generalization challenge, qualitatively different from standard random splits (Section 4.1). **Depth-Driven Generalization:** We then investigate the source of this challenge. Through a controlled scaling study in Section 4.2, we demonstrate that generalization performance is primarily driven by the functional diversity (or evolutionary depth) of the training data. **Internal Consistency and Robustness:** Finally, to confirm the integrity of our benchmark, we demonstrate its large-scale consistency in Section 4.3 by experimentally validating both the fidelity of its gold-standard biological assays and its computational robustness.

## 4.1 OUT-OF-DISTRIBUTION CHALLENGE: THE TEMPORAL SPLIT

To accurately reflect the challenges inherent in a real-world protein-directed evolution process, our benchmark incorporates a temporal data split, specifically designed to assess how well models generalize across an evolutionary trajectory. Unlike traditional benchmarks that rely on static, random data splits, this approach simulates the iterative nature of protein engineering.

### 4.1.1 EXPERIMENTAL SETTINGS

**Dataset** We emulate a practical protein engineering scenario by splitting the data temporally: rounds 1–27 for training, 28 for validation, and 29–31 for testing. The nucleic acid (DNA/RNA) dataset contains 729,302 training sequences, 148,014 validation sequences, and 149,884 test sequences. The protein dataset comprises 256,429 training sequences, 45,208 validation sequences, and 108,232 test sequences. Crucially, our novel high-throughput wet-lab platform (in Section B) allows TadABench-1M to be a continually evolving resource. As new rounds of data are generated, the current test set can be integrated into the training data, and the new rounds will serve as a fresh, unseen test set. This approach guarantees that the benchmark remains a realistic and challenging OOD task over time. To this end, TadABench-1M is designed for ongoing expansion as a challenging resource.

**Task and Evaluation** The primary task is to predict the relative fitness of unseen protein variants, a core challenge in engineering workflows. We evaluate models on their ability to rank variants correctly using three standard metrics (Notin et al., 2024): Spearman's rank correlation ($\rho$) to assess overall monotonic trend prediction, Recall@10% to measure the identification rate of top-tier candidates, and normalized Discounted Cumulative Gain at 10% (nDCG@10%) to evaluate the ranking quality within this top decile, offering a multi-faceted view of their performance.

**Models** We evaluate a suite of biological language models (BLMs) spanning DNA, RNA, and protein domains. For DNA, we include models from the EVO2 (Brixi et al., 2025) and Nucleotide-Transformer (NT) (Dalla-Torre et al., 2023) families. For RNA, we test the OmniGenome (OG) family (Yang & Li, 2024). For proteins, we consider the ESM2 (Lin et al., 2023), ProtTrans (Elnaggar et al., 2021), and ESMC (ESM Team, 2024) families. It is important to note that due to the modality differences (nucleic acid vs. protein sequences), direct performance comparisons between these model classes are not meaningful. Details on experimental settings are provided in Section D.1.

### 4.1.2 RANDOM SPLIT: TADABENCH-1M IS LEARNABLE

To establish a baseline and verify that the fitness landscape of TadABench-1M is inherently learnable, we first evaluated models under an ideal in-distribution (i.i.d.) scenario using a standard random 8:1:1 split (train:validation:test). As shown in Table 1, BLMs perform remarkably well under these conditions, achieving a Spearman correlation of approximately $\rho \approx 0.8$ on both the validation and test sets. This strong performance confirms that the sequence-activity relationships within the benchmark are coherent and effectively learnable in an i.i.d. setting. The resulting high-performance baseline of $\rho \approx 0.8$ thus serves as a crucial reference point, validating the benchmark's fundamental learnability and setting the stage for evaluating model generalization on more challenging splits.

Table 1: Performance on TadABench-1M (protein version). The training, validation, and test sets were obtained via an 8:1:1 random split. For each model, the best result is selected from three different learning rates. **Bold** numbers indicate the highest performance.

| Model | Validation | | | Test | | |
|---|---|---|---|---|---|---|
| | Spearman | Recall@10% | nDCG@10% | Spearman | Recall@10% | nDCG@10% |
| ESM2-35M | 0.8032 | 0.1830 | 0.4824 | 0.8014 | 0.1617 | 0.4814 |
| ESM2-150M | 0.7386 | 0.2290 | 0.4364 | 0.7371 | 0.2324 | 0.4437 |
| ESM2-650M | 0.5360 | 0.1793 | 0.4740 | 0.5348 | 0.1710 | 0.4779 |
| Prot-BERT | 0.7910 | 0.2230 | 0.4879 | 0.7883 | 0.2262 | 0.4918 |
| Prot-XLNET | 0.8054 | 0.2264 | 0.4912 | 0.8030 | 0.2193 | 0.4965 |
| ESMC-300M | 0.8102 | 0.2439 | 0.4959 | 0.8067 | **0.2363** | **0.4995** |
| ESMC-600M | **0.8127** | **0.2446** | **0.5006** | **0.8079** | 0.2317 | 0.4949 |

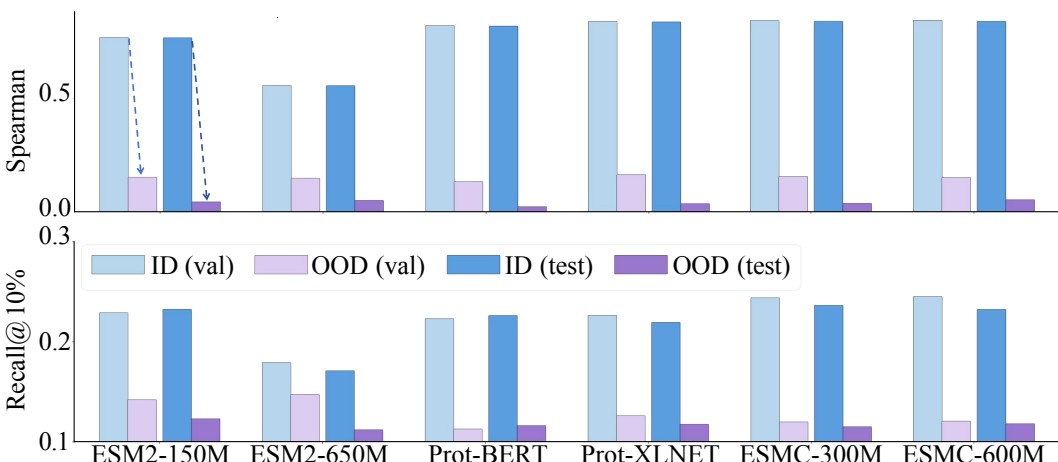

Figure 4: The temporal split of our TadABench-1M, designed to simulate real-world TadA engineering scenarios, presents a challenging task for protein language models. In this context, "ID" refers to the in-distribution setting, represented by the random split, while "OOD" corresponds to the out-of-distribution setting, indicated by the temporal split. The shallow color bars denote the validation set, while the deeper color bars represent the test set.

### 4.1.3 TEMPORAL SPLIT: TADABENCH-1M IS CHALLENGING

**In stark contrast, when evaluated on the practical temporal split, all BLMs exhibit a dramatic drop in performance as illustrated in Figure 4.** As shown in Table 3 and Table 2, the Spearman correlation for linear probing collapses to approximately $\rho \approx 0.1$. This severe performance degradation highlights a critical generalization gap. Even a simple one-hot encoding baseline fails, yielding correlations of 0.0707 on the validation set and 0.0459 on the test set, underscoring the profound OOD difficulty. These results highlight the difficulty of practical OOD scenarios.

**Fine-tuning and prompt tuning do not overcome the generalization gap.** To investigate if performance could be improved by training more parameters, we evaluated both full fine-tuning and prompt tuning. As shown in Table 5, fine-tuning yielded no significant improvement over linear probing. Similarly, an extensive search over prompt tuning hyperparameters—including varying prompt length, position, initialization, and layer—also failed to lift the Spearman correlation above the $\rho \approx 0.1$ (Table 6). This suggests the bottleneck is a more fundamental generalization challenge.

Table 2: Performance on TadABench-1M (DNA version). Data from rounds 1–27 is used for training, round 28 for validation, and rounds 29–31 for testing. For each model, the best result is selected from three different learning rates. The **Bold** numbers indicate the highest performance. We here report the RNA language model, OmniGenome (OG), since the dataset is the same after changing T to U.

| Model | Validation | | | Test | | |
|---|---|---|---|---|---|---|
| | Spearman | Recall@10% | nDCG@10% | Spearman | Recall@10% | nDCG@10% |
| Evo-7B | 0.0490 | 0.1097 | 0.2604 | **0.0707** | 0.1005 | 0.3236 |
| Evo-40B | **0.0980** | **0.1157** | **0.2702** | 0.0675 | 0.1003 | **0.3244** |
| NT-50M | 0.0401 | 0.0959 | 0.2464 | 0.0166 | 0.0950 | 0.3109 |
| NT-100M | 0.0520 | 0.0982 | 0.2485 | 0.0045 | 0.0870 | 0.3048 |
| NT-250M | 0.0470 | 0.0858 | 0.2137 | 0.0006 | 0.0971 | 0.3085 |
| NT-500M | 0.0361 | 0.0985 | 0.2225 | 0.0189 | 0.1005 | 0.3079 |
| OG-46M | 0.0555 | 0.0911 | 0.2192 | 0.0079 | **0.1063** | 0.3158 |
| OG-418M | 0.0078 | 0.0949 | 0.2391 | 0.0048 | 0.0859 | 0.3042 |

Table 3: Performance on TadABench-1M (protein version). Data from rounds 1–27 is used for training, round 28 for validation, and rounds 29–31 for testing. For each model, the best result is selected from three different learning rates. **Bold** numbers indicate the highest performance.

| Model | Validation | | | Test | | |
|---|---|---|---|---|---|---|
| | Spearman | Recall@10% | nDCG@10% | Spearman | Recall@10% | nDCG@10% |
| ESM2-150M | 0.1458 | 0.1420 | 0.6569 | 0.0416 | **0.1230** | **0.3068** |
| ESM2-650M | 0.1423 | **0.1473** | 0.6530 | 0.0479 | 0.1120 | 0.2791 |
| Prot-BERT | 0.1280 | 0.1128 | 0.6534 | 0.0214 | 0.1162 | 0.2980 |
| Prot-XLNET | **0.1570** | 0.1261 | **0.6589** | 0.0342 | 0.1175 | 0.2895 |
| ESMC-300M | 0.1498 | 0.1199 | 0.6495 | 0.0355 | 0.1151 | 0.2867 |
| ESMC-600M | 0.1452 | 0.1206 | 0.6397 | **0.0509** | 0.1180 | 0.2860 |

## 4.2 GENERALIZATION IS DRIVEN BY EVOLUTIONARY DEPTH

Building upon the established OOD challenge, this section investigates its underlying causes. We hypothesize that the generalization gap arises not from the quantity of training data, but from its functional diversity. In this context, functional diversity arises directly from the benchmark's inherent evolutionary depth, which spans multiple rounds of mutation and selection. To test our hypothesis, we conduct a controlled scaling study where we vary the training set composition while keeping the validation and test sets fixed. We partition the training data using three distinct strategies:

**Density:** A random subsample of the full training set. This simulates simply collecting more data from the same evolutionary rounds.

**Diversity:** Sequences selected from all training rounds to maximize similarity with the validation set. This prioritizes data functionally closer to the target.

**Round:** Entire experimental rounds selected based on their aggregate similarity to the validation set. This preserves intra-round correlations while moving closer to the target task.

As shown in Figure 5, performance scales with the training data size (x-axis, log scale), but the scaling law is entirely dependent on the curation strategy. For all tested models, the *diversity* and *round*-based strategies consistently and significantly outperform the naive *density*-based strategy. For instance, a small, diversity-focused dataset often yields better performance than a much larger but randomly sampled one. This strongly indicates that successfully bridging the OOD gap requires exploring functionally diverse regions of the sequence space, rather than simply increasing density.

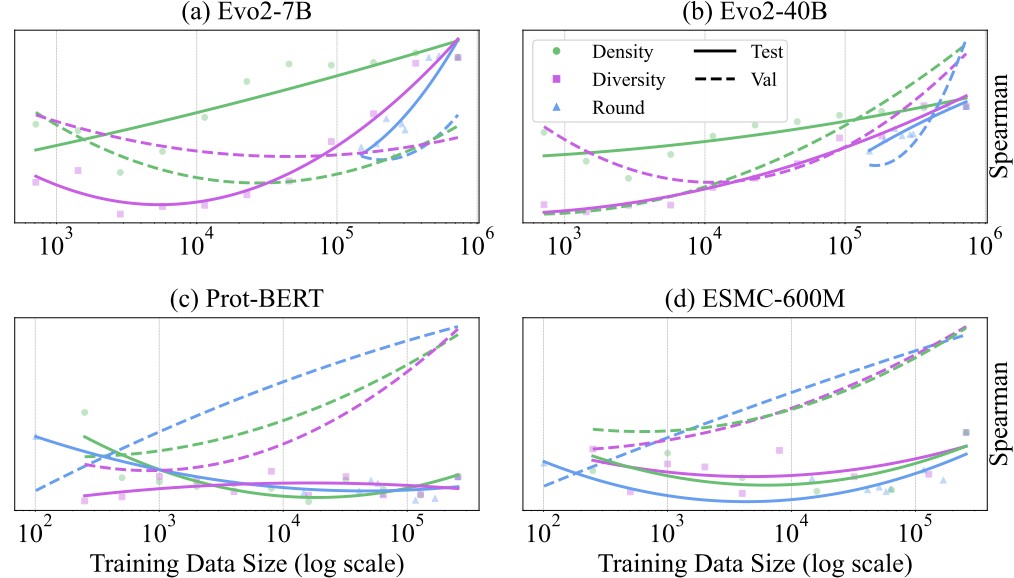

Figure 5: Performance by data scales in different modes of segmentation on diverse models. The rough data scaling trend can be observed among all models, especially on the domain. Note that the dataset of the DNA version (a, b) and protein version (c, d) is different.

### 4.3 IN VITRO AND IN SILICO DATA CONSISTENCY

The consistency of TadABench-1M is confirmed through two key validations: the accuracy of the in vitro sequencing ranks and the robustness of the in silico Seq2Graph computation. This rigorous validation ensures that the inherent OOD challenge are both genuine and reliable.

For wet experiments, we performed orthogonal validation using a low-throughput, gold-standard assay to ensure that our high-throughput fitness labels are biologically meaningful. We engineered a Green Fluorescent Protein (GFP) reporter system to reflect the protein activity. We selected several key variants spanning a range of fitness score and independently measured their activity using this GFP assay. The results provided strong validation: the rank ordering of variant activities determined by the GFP assay showed an exceptionally high correlation with the rank ordering derived from our high-throughput NGS read counts enrichment.

For dry experiments, we conduct two bootstrapping analyses to evaluate the robustness of Seq2Graph. First, we randomly subsample 50% of the experimental rounds (15 of 31) and rerun the entire construction pipeline. The resulting activity scores for sequences common to both datasets show a Spearman's rank correlation of $\rho = 0.90$ ($p < 10^{-5}$). Second, we subsample 50% of sequences within each round before running the pipeline, which yields a Spearman's $\rho = 0.95$ ($p < 10^{-5}$). These high correlations demonstrate that our method is robust to variations in experimental input and sequencing depth. Together, these in vitro and in silico validations confirm the internal consistency.

## 5 CONCLUSION

In this study, we introduce TadABench-1M, a benchmark designed to evaluate Biological Language Models (BLMs) in out-of-distribution (OOD) settings, using the engineering of the TadA protein as a case study. By incorporating temporal splits and evolutionary depth, we simulate real-world challenges such as generalizing to unseen sequences, extrapolating across iterative mutations, and ensuring consistent fitness measurements. Our experiments reveal a significant performance gap, with state-of-the-art BLMs failing to generalize under realistic OOD conditions despite excelling in standard IID evaluations. This work emphasizes the importance of evolutionary depth over sheer data volume for OOD generalization and provides key insights for designing future benchmarks.

## ETHICS STATEMENT

This research adheres to ethical guidelines set forth by the ICLR Code of Ethics. The work presented involves no human subjects, animals, or sensitive data that would require ethical approval from an institutional review board (IRB). All data used in this study were derived from wet-lab experiments in which proper ethical standards were followed. The authors declare no conflicts of interest or external sponsorship that may have influenced the research outcomes.

The dataset used in this study is built from over one million variants of the TadA enzyme, generated through iterative wet-lab processes. All data collection methods have been carried out in compliance with standard scientific practices, and the privacy and security of the biological data have been ensured. The study does not present any harmful insights or methodologies that could lead to discriminatory, biased, or unethical applications in protein engineering or other fields.

## REPRODUCIBILITY STATEMENT

Detailed descriptions of the wet-lab experimental procedures used to generate the data, including all relevant protocols, can be found in the Section B. To ensure the reproducibility of our research, we have made all relevant materials available in the supplementary materials. A comprehensive README file accompanies the code to guide users in reproducing the results. We encourage others to follow these guidelines to replicate the experiments and verify the conclusions presented in this paper. All data processing steps, model training, and evaluation procedures are transparently documented to facilitate reproducibility. The dataset used in this study, along with the code for processing and analyzing the data, will be made accessible upon acceptance.

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

Figure 6: TadABench-1M is derived from extensive wet-lab protein evolution experiments encompassing 31 iterative rounds. *Left* panel illustrates the functional role of TadA as a key enzyme in base editing. *Central* panel presents the in vitro evolution process of TadA, which enabled the generation of comprehensive deep sequencing data. *Right* panel outlines the construction of our dataset using the Seq2Graph method, followed by its evaluation with biological language models.

## APPENDIX

We first discuss the use of Large Language Models in Section A. Additionally, we introduce our wet-lab data collection framework in Section B. Section C provides biological details on dataset construction, while Section D presents additional experimental settings and results.

## A    THE USE OF LARGE LANGUAGE MODELS

During the preparation of this paper, large language models (LLMs) such as GPT-5 (OpenAI, 2025) were utilized to assist with various tasks, including grammar and error correction, enhancing wording and phrasing, and providing recommendations for the visualization of tables and figures. These tools primarily contributed to improving the linguistic quality of the manuscript and offering suggestions for more effective visual representation of data. However, the LLMs were not involved in the scientific ideation or the development of research concepts. Their contributions are disclosed here to ensure transparency in the research methodology.

## B    BIOLOGICAL ASSAY

We provide a visual summary of our entire pipeline, from the biological experiments to the final benchmark construction, illustrated in Figure 6.

### B.1    BIOLOGICAL AND EXPERIMENTAL PRELIMINARIES

**Adenine Base Editing and TadA:**    Base editing enables precise single-nucleotide conversions without the double-strand breaks (DSBs) associated with traditional CRISPR methods, offering improved safety and precision (Cox et al., 2015; Hilton & Gersbach, 2015; Komor et al., 2016; Gaudelli et al., 2017). Adenine base editors (ABEs), the focus of this work, convert A•T to G•C base pairs. This is achieved using an engineered tRNA-specific adenosine deaminase (TadA) enzyme, which catalyzes the deamination of adenosine (A) to inosine (I), an intermediate interpreted as guanosine (G) during DNA replication. Our work centers on TadA8e (Richter et al., 2020), a high-activity variant comprising 167 amino acids. Our benchmark, TadABench-1M, is constructed from libraries of TadA8e variants generated through extensive mutagenesis.

**PANCE for High-Throughput Activity Annotation:**    We employ Phage-Assisted Non-Continuous Evolution (PANCE) (Miller et al., 2020; Zhang et al., 2024), a directed evolution method, to screen TadA variants for their activity at scale. In this system, a library of TadA variants is encoded within a bacteriophage population. Phages carrying variants with higher enzymatic activity replicate more

efficiently, leading to their enrichment. The relative abundance of each variant is then quantified via Next-Generation Sequencing (NGS), where higher read counts serve as a proxy for superior activity. Our TadABench-1M dataset comprises data from 31 independent PANCE experiments, each screening a unique library of tens of thousands of TadA variants.

**Degenerate Sequences for Efficient Library Synthesis:**   Synthesizing large and diverse variant libraries by creating each DNA sequence individually is prohibitively expensive. To overcome this, we utilize degenerate sequence synthesis (Li et al., 2022). A degenerate sequence is a single oligonucleotide template that contains ambiguous nucleotide codes (*e.g.*, 'N' representing A, C, G, or T) at specified positions. This technique enables the cost-effective generation of a vast library containing thousands of unique variants from a single synthesis reaction, which was essential for constructing the diverse libraries screened in our PANCE experiments.

### B.2 Experimental Workflow for Large-Scale Protein Fitness Dataset

Out-of-distribution (OOD) data is frequently encountered in real-world applications, such as protein engineering. Our objective was to generate an extensive protein fitness dataset to evaluate the OOD generalization of biological language models rigorously. Therefore, it required a scalable and generalizable experimental workflow to map a sequence-function landscape far from naturally occurring proteins.

We developed a general multi-stage workflow centered on Phage-Assisted Non-Continuous Evolution (PANCE), which integrates expert-guided library design, high-throughput selection, and orthogonal validation. The efficacy of this workflow was demonstrated by its primary outcome: the discovery of a state-of-the-art TadA variant whose activity significantly surpasses published benchmarks (Richter et al., 2020). This result validates the biological relevance of our high-throughput fitness measurements and supports the integrity of the generated dataset.

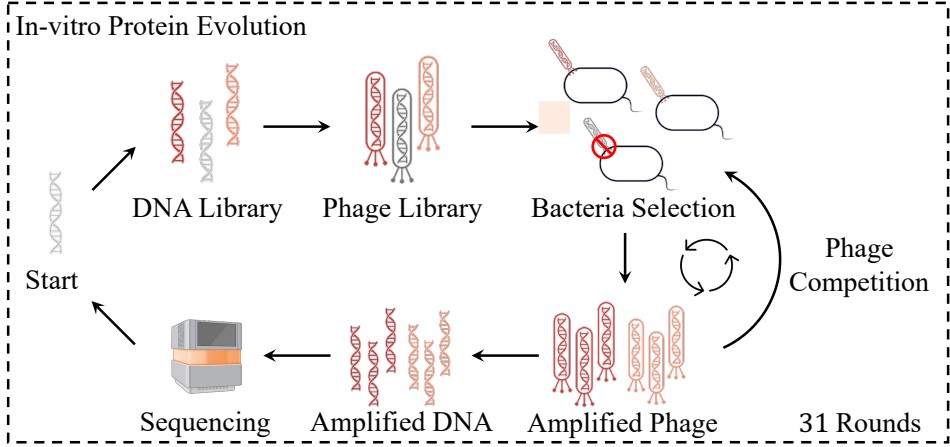

Figure 7: The workflow for general large-scale protein activity annotation. A large library of mutants is synthesized by degenerate sequences, and activity candidates are labeled by phage-assisted evolution. Variants with higher activity trigger gIII expression, leading to phage propagation.

### B.2.1 Expert-Guided Library Design

A key departure from standard continuous evolution systems that rely on random mutagenesis is our use of expert-designed libraries. To ensure the utility and generality of the OOD data, we adopted a well-established pipeline common in industrial protein engineering. Each of the 31 experimental rounds began with a unique, rationally designed library of TadA variants. Protein engineers, leveraging structural information and domain expertise, identified promising regions for mutation and designed targeted libraries using degenerate sequence synthesis (*e.g.*, using NNK codons). This approach serves two critical functions:

**Reflects Applied Engineering Practices:** This approach introduces a knowledge-driven bias, focusing exploration on promising regions of the sequence space. This strategy mirrors common industrial practice, which favors targeted mutagenesis over uniformly random approaches.

**Establishes a Controlled Baseline:** Degenerate synthesis ensures that all designed variants are present at a uniform initial concentration. This experimental control is critical: by eliminating initial abundance as a confounding variable, it greatly simplifies downstream fitness estimation.

### B.2.2 PHAGE-ASSISTED NON-CONTINUOUS EVOLUTION (PANCE)

We employed Phage-Assisted Non-Continuous Evolution (PANCE) (Miller et al., 2020; Zhang et al., 2024) as our high-throughput selection engine. The core principle of PANCE is coupling the function of a target protein to the replication of a bacteriophage. In our system, we engineered the M13 phage by deleting the essential gene *gIII*, which encodes the pIII protein required for virion release. The expression of *gIII* was placed under the control of a reporter system activated by the enzymatic activity of our target protein, TadA. Consequently, only *E. coli* cells hosting a sufficiently active TadA variant can produce and release new phages.

To ensure a robust selection process, each of the 31 rounds involved five cycles of serial dilution and competitive regrowth. This iterative enrichment minimizes stochastic effects and ensures the final phage population's composition accurately reflects the relative fitness of the encoded variants. Phages encoding high-activity proteins replicate more efficiently, ultimately dominating the population, while those encoding low-activity variants are progressively eliminated.

### B.2.3 FITNESS QUANTIFICATION AND ROBUSTNESS

We quantified the fitness of each TadA variant based on its read count enrichment, as determined by deep Next-Generation Sequencing (NGS) of the enriched phage population. To guarantee precision and minimize sampling noise, each round was sequenced at an exceptional depth of over 100G bases, ensuring accurate quantification of variant frequencies.

### B.2.4 ORTHOGONAL VALIDATION OF FITNESS LABELS

To address the critical question of whether our high-throughput fitness labels are biologically meaningful, we performed rigorous orthogonal validation using a low-throughput, gold-standard assay. We engineered a Green Fluorescent Protein (GFP) reporter system in which a mutated, non-fluorescent GFP gene could be repaired by the editing activity of a TadA variant, leading to a measurable fluorescent signal.

We selected several key variants spanning a range of fitness scores from the PANCE screens and independently measured their activity using this GFP assay. The results provided strong validation: the rank ordering of variant activities determined by the GFP assay showed an exceptionally high correlation (*e.g.*, Spearman's rank correlation $\rho > 0.99$) with the rank ordering derived from our high-throughput NGS read counts enrichment. This confirmation anchors our large-scale dataset in established biophysical measurements and affirms that the fitness labels in TadABench-1M are reliable and biologically meaningful.

*We are unable to release the full, detailed experimental protocols due to company licensing restrictions.*

## C DETAILS OF SEQ2GRAPH

This section provides further biological and methodological details on the construction of Seq2Graph, complementing the computational overview presented in Section 3.

## C.1 DIRECTED GRAPH CONSTRUCTION

In our directed evolution experiments, the activity of a TadA variant is proportional to its replication rate, which is measured by read counts from Next-Generation Sequencing (NGS)[3]. A common practice is to normalize these counts to estimate variant activities (Wagner et al., 2012; Love et al., 2014; Robinson & Oshlack, 2010). However, while our 31 experimental rounds followed a standardized protocol, inherent biological and technical stochasticity introduces batch effects, making it impossible to aggregate normalized counts from different rounds into a single, consistent dataset.

To solve such limitations, we propose to model the relative activity of variants in a directed graph, rather than the common practice of absolute activity after normalization, shown in Figure 3 (a). In the directed graph, $G = (V, E)$, the DNA sequence of each variant obtained from NGS is taken as a node $v_i$. For the list of growth multiples for read counts, the number of edges in $G$ can be up to $|V|^2$. Considering the computational complexity, We require a sparse graph that remains weakly connected, as this property is essential for the relative activity assignment.

Specifically, we sort the list of growth multiples for read counts and only add edges for nodes with adjacent count values along the list. To further manage computational complexity, we cap the number of edges generated per experiment at 100,000. Each edge points from nodes with higher activity to those with lower activity, which means the edge weight is always greater than 1. The weight of edge $e_{i \to j}$ represents the relative activity of $v_i$ over $v_j$, $w_{ij} = \frac{C(v_i)}{C(v_j)}$, where $C(\cdot)$ is the growth multiples for read counts.

Since the directed evolution process iteratively enriches for high-performing variants, some sequences are present across multiple experimental rounds, acting as bridging nodes within the dataset. Consequently, the graph composed of all 31 experimental datasets is weakly connected, ensuring a relative activity path exists between nearly any two variants. We also store the experimental round as a node attribute; on average, each node appears in 1.58 rounds.

## C.2 INCONSISTENCY ELIMINATION

Aggregating pairwise comparisons from 31 rounds inevitably introduces conflicting relationships (*e.g.*, $v_i > v_j$, $v_j > v_k$, but $v_k > v_i$). These conflicts manifest as cycles, or strongly connected components (SCCs), in the graph, as shown in Figure 3 (b). To ensure a globally consistent activity ranking, these cycles must be eliminated.

We assume that edges with higher weights, corresponding to larger read count ratios, are more reliable as they are less susceptible to measurement noise. We therefore frame the cycle removal task as the weighted Feedback Arc Set (FAS) problem: finding a minimum-weight set of edges whose removal makes the graph acyclic, shown in Equation (2).

$$\min_{F \subseteq E} \quad \sum_{e \in F} w_e \tag{2}$$
$$\text{s.t.} \quad G' = (V, E \setminus F) \text{ is acyclic}$$

As the exact FAS problem is NP-hard and our graph is large, we employ a fast greedy heuristic (Eades et al., 1993) that approximates the solution by iteratively ordering vertices based on their in- and out-degrees. For efficiency, we apply this algorithm independently to each SCC. The result is a directed acyclic graph (DAG) containing the original set of nodes but with fewer edges.

## C.3 ACTIVITY ASSIGNMENT

With the DAG established, we assign a consistent, relative activity score to each node. We anchor the scale by setting the activity of the reference sequence (TadA8e (Richter et al., 2020)) to 1.0. Since phage replication exhibits exponential growth, we propagate activity scores in a log-additive fashion along the graph edges.

---

[3]Next-Generation Sequencing (NGS) is a technology platform that enables high-throughput sequencing. In this paper, we use NGS and deep sequencing interchangeably.

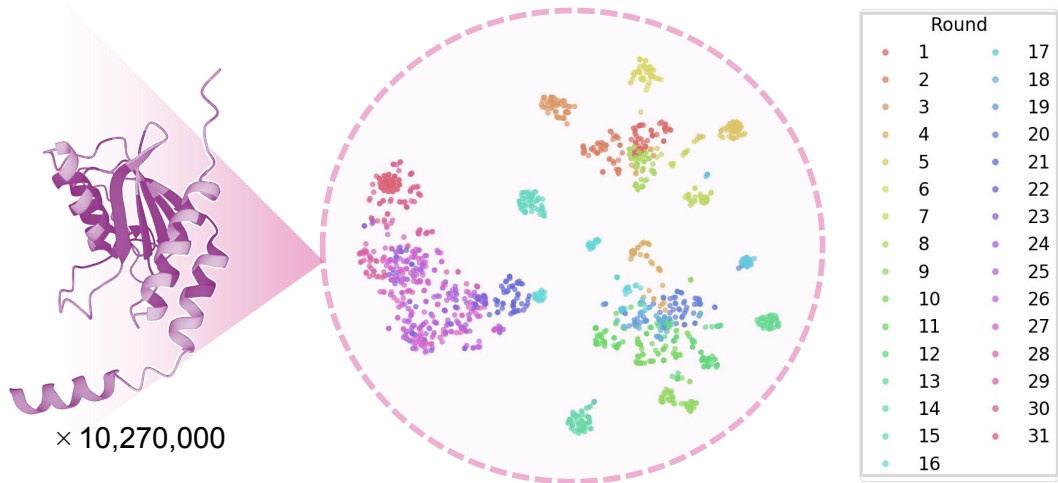

× 10,270,000

Figure 8: Visualization of our dataset of TadABench-1M. *Left* part shows the folded structure of TadA obtained from our wet-lab experiments, predicted using ESMFold (Lin et al., 2023). *Right* part presents a t-SNE visualization of the clustering results across 31 rounds of protein evolution.

We traverse the graph from the reference node using a breadth-first search (BFS) to assign activities. We chose BFS because it identifies the shortest path from a source to all other nodes. Shorter paths involve fewer pairwise comparisons and are therefore less likely to accumulate experimental noise, yielding the highest-confidence estimates. To implement this, the graph is treated as undirected during the traversal. A node's activity is calculated and permanently set upon its first visit, which ensures the calculation is based exclusively on a single shortest path. This process yields the DNA version of Seq2Graph, which contains 1,027,200 DNA sequences with their assigned activity labels.

Finally, to create a protein-level dataset, we map DNA-level activities to their corresponding protein sequences. Since multiple DNA variants can encode the same protein, we compute the final activity for each protein by averaging the activities of all its corresponding DNA sequences. This averaging is non-trivial, as synonymous DNA variants can exhibit different activities due to effects like codon usage bias. This final step produces our protein dataset, comprising 409,869 annotated protein sequences.

Based on Seq2Graph, we construct the final dataset, visualized in Figure 8. The t-SNE embedding on the right, generated from 50 sequences sampled from each of the 31 rounds, reveals distinct clusters corresponding to different rounds. It depicts that our multi-round experimental design explores diverse regions of the sequence space, providing a broad dataset for a general evaluation.

## C.4 THE CAS9 DATASET BUILT BY SEQ2GRAPH

The Seq2Graph framework is a task-agnostic and generalizable method. Its core principle revolves around reconciling locally-consistent rankings through a graph-based approach, which is inherently not tied to any single protein system or specific dataset. This makes Seq2Graph applicable to a broad range of tasks. Whether it involves large-scale labels across multiple independent experiments, batches, or even different laboratories, the framework remains robust and versatile.

To demonstrate the generality and versatility of the Seq2Graph framework, we extend its application to the creation of a Cas9 dataset, as presented in Table 4. This showcases the adaptability of Seq2Graph in generating high-quality data representations across diverse domains. To further illustrate the power of our approach, we provide a visualization of the generated labels in Figure 9, which highlights the distribution of values across different modalities. This demonstrates the framework's flexibility in handling various biological data types, reinforcing its utility across different scientific domains and tasks.

Table 4: Statistics of the Cas9 dataset created by Seq2Graph. The dataset is split into training, validation, and test sets with a ratio of 8:1:1.

|         | Train  | Val   | Test  | Total  |
|---------|--------|-------|-------|--------|
| DNA     | 255918 | 31989 | 31991 | 319898 |
| RNA     | 255918 | 31989 | 31991 | 319898 |
| Protein | 131735 | 16466 | 16468 | 164669 |

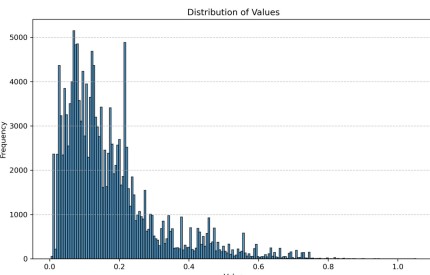 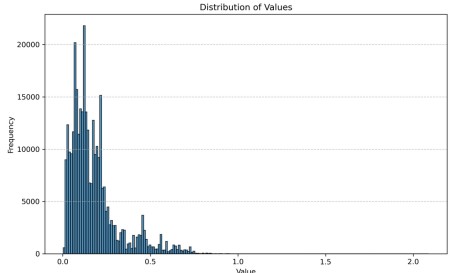

Figure 9: The value distribution of the Cas9 dataset created by Seq2Graph. The *left* figure represents the activity scores in the DNA/RNA modality, while the *right* figure corresponds to the protein modality.

## D EXPERIMENTAL DETAILS OF TADABENCH-1M

### D.1 DATASET, TASK, AND EVALUATION

**Dataset and Task.** TadABench-1M is derived from Next-Generation Sequencing (NGS) data of evolved TadA variants, available as both nucleic acid (DNA/RNA) and protein sequences (see Section 3.4). Our primary task mimics a realistic directed evolution scenario where models must predict the relative fitness of novel variants. Success is measured not by precise regression of activity values, but by the ability to correctly rank top-performing sequences. This is because experimental validation capacity is limited, making the accurate identification of a small set of promising candidates paramount.

**Data Splits.** We employ two distinct data-splitting strategies to evaluate model performance under different conditions.

- **Practical (Temporal) Split:** To simulate a real-world, forward-in-time prediction scenario, we split the data temporally: rounds 1–27 for training, round 28 for validation, and rounds 29–31 for testing. The nucleic acid dataset contains 729,302 training, 148,014 validation, and 149,884 test sequences. The protein dataset comprises 256,429 training, 45,208 validation, and 108,232 test sequences.

- **Random (ID) Split:** To establish an idealized baseline, we create a standard independent and identically distributed (ID) split by randomly partitioning the entire dataset into training (80%), validation (10%), and test (10%) sets.

**Evaluation Metrics.** Following established practices in protein fitness prediction (Notin et al., 2024), we evaluate models using three key metrics. **Spearman's rank correlation coefficient ($\rho$)** measures the model's ability to capture the overall monotonic relationship between predicted and true activity scores. To assess performance on identifying top candidates, we use **Recall@10%**, the fraction of true top-10% variants identified in the predicted top 10%, and **normalized Discounted Cumulative Gain at 10% (nDCG@10%)**, which further evaluates the ranking quality within this top decile.

Table 5: Performance on TadABench-1M using fine-tuning. Data from rounds 1–27 is used for training, round 28 for validation, and rounds 29–31 for testing.

| Model | Learning Rate | Spearman | Recall@10% | nDCG@10% |
|---|---|---|---|---|
| | 3e-5 | 0.0432 | 0.1260 | 0.3094 |
| ESM2-8M | 1e-4 | 0.0553 | 0.1270 | 0.3066 |
| | 3e-4 | 0.0407 | 0.1000 | 0.2594 |
| | 3e-5 | 0.0465 | 0.1136 | 0.2828 |
| ESM2-35M | 1e-4 | 0.0479 | 0.1157 | 0.2817 |
| | 3e-4 | 0.0072 | 0.1033 | 0.2602 |
| | 3e-5 | 0.0271 | 0.1418 | 0.3203 |
| ESM2-150M | 1e-4 | 0.0391 | 0.1127 | 0.2784 |
| | 3e-4 | 0.0271 | 0.1418 | 0.3203 |
| | 3e-5 | 0.0484 | 0.0899 | 0.3119 |
| NT-50M | 1e-4 | 0.0296 | 0.0850 | 0.3055 |
| | 3e-4 | 0.0491 | 0.0848 | 0.3063 |
| | 3e-5 | 0.0367 | 0.0871 | 0.3083 |
| NT-100M | 1e-4 | 0.0460 | 0.0887 | 0.3130 |
| | 3e-4 | 0.0480 | 0.0907 | 0.3106 |
| | 3e-5 | 0.0368 | 0.0841 | 0.3077 |
| NT-250M | 1e-4 | 0.0237 | 0.0840 | 0.3068 |
| | 3e-4 | 0.0485 | 0.0862 | 0.3080 |
| | 3e-5 | 0.0630 | 0.0913 | 0.3147 |
| NT-500M | 1e-4 | 0.0465 | 0.0853 | 0.3064 |
| | 3e-4 | 0.0492 | 0.0888 | 0.3117 |

## D.2 MODELS AND IMPLEMENTATION DETAILS

**Pre-trained Models.** We evaluate a diverse suite of pre-trained biological language models (BLMs). For nucleic acids, we test models from the EVO2 (Brixi et al., 2025) family (EVO2-7B, EVO2-40B) and NucleotideTransformer (Dalla-Torre et al., 2023) family (NT-50M, NT-100M, NT-250M, NT-500M). For proteins, we include models from ESM2 (Lin et al., 2023) (ESM2-35M, ESM2-150M, ESM2-650M), ProtTrans (Elnaggar et al., 2021) (Prot-BERT, Prot-XLNET), and ESMC (ESM Team, 2024) (ESMC-300M, ESMC-600M). When using RNA-specific models, DNA sequences were processed by mapping Thymine (T) to Uracil (U).

**Linear Probing Implementation.** For our primary evaluation, we perform linear probing on features extracted from pre-trained models. For most models, we use the final hidden state representation; for EVO2 models, we use the output logits. To ensure a fair comparison across models with different embedding dimensions, we train a two-layer MLP regression head on top of these features. The hidden layer size of the MLP is adjusted for each model to maintain a consistent number of trainable parameters, with a ReLU activation between layers. We found that using all logit dimensions for EVO2 models led to training instability. We therefore stabilized training by using only the logits corresponding to the four canonical nucleotides (A, T, C, G). For all models, we use the sequence of token representations as input to the MLP, as this performed comparably to mean-pooling while retaining more information. Each head is trained for 20 epochs using a cosine learning rate scheduler with a 1-epoch warmup, selecting the best learning rate from {3e-5, 1e-4, 3e-4} based on validation performance.

## D.3 PERFORMANCE ON THE PRACTICAL (TEMPORAL) SPLIT

To assess whether training a larger number of parameters could bridge the generalization gap observed with linear probing, we evaluated both full model fine-tuning and prompt tuning. As shown in Table 5, fine-tuning all model parameters provided no significant improvement over linear probing. Similarly,

Table 6: Performance on TadABench-1M using prompt tuning with different hyper-parameters. Data from rounds 1–27 is used for training, round 28 for validation, and rounds 29–31 for testing.

| Model | Prompt Init | Prompt Position | Prompt Length | Prompt Layers | Test Spearman |
|-------|-------------|-----------------|---------------|---------------|---------------|
| NT-50M | kaiming | add | 1 | 0 | 0.0405 |
| | uniform | add | 1 | 0 | 0.0683 |
| | uniform | prepend | 1 | 0 | 0.0617 |
| | uniform | prepend | 2 | 0 | 0.0653 |
| | uniform | prepend | 4 | 0 | 0.0652 |
| | uniform | prepend | 8 | 0 | 0.0735 |
| | uniform | prepend | 16 | 0 | 0.0656 |
| | uniform | prepend | 32 | 0 | 0.0754 |
| | uniform | prepend | 64 | 0 | 0.0606 |
| ESMC-300M | uniform | prepend | 1 | 0 | 0.0182 |
| | uniform | prepend | 1 | 10 | 0.0266 |
| | uniform | prepend | 1 | 20 | 0.0239 |
| | uniform | prepend | 1 | 30 | 0.0193 |

despite an extensive hyperparameter search for prompt tuning (including prompt length, position, and initialization), performance remained poor, with Spearman correlation failing to exceed $\rho \approx 0.1$, as shown in Table 6. These results suggest that the performance bottleneck is not the number of trainable parameters but rather a more fundamental generalization failure of the pre-trained representations on this out-of-distribution task.

