# OpenReview forum: "TadABench-1M: A Large-Scale Wet-Lab Protein Benchmark For Rigorous OOD Evaluation"
_ICLR.cc/2026/Conference — Submitted to ICLR 2026_

### Official Review · Reviewer_ZqXN · 2025-10-27

**Soundness:** 2
**Presentation:** 2
**Contribution:** 2
**Rating:** 4
**Confidence:** 3

**Summary:**

This paper proposes TadABench-1M, a benchmark based on a wet-lab dataset of over one million variants of the therapeutically relevant TadA enzyme, aiming to focus on realistic out-of-distribution cases, evolutionary depth, and consistency in measurement. This paper also introduces Seq2Graph, a scalable graph-based algorithm that systematically unifies multi-batch experimental data. A key finding in this work is that while state-of-the-art biological language models excel on a standard random split of the data, they fail dramatically on a realistic temporal prediction task.

**Strengths:**

- The temporal split of the dataset in this benchmark establishes a realistic evaluation setting that mimics a real-world engineering campaign, and also reveals a significant generalization gap.
- This work demonstrates that sequence diversity and evolutionary depth are more critical for OOD generalization than raw data volume

**Weaknesses:**

- The benchmark is designed around TadA enzyme and it’s evolution. This setting may restrict the generalization to other protein families. Also, it’s not clear whether the findings in this benchmark can be transferred to other proteins.
- This paper mainly focuses on biological language models, such as ESM, Evo, etc. Is it possible that the findings in this work only apply to language-based methods, it would be interesting to see structure-aware methods also included in the benchmark.

**Questions:**

Please refer to weakness section

---

> ### Author Response · Authors · 2025-11-28
>
> We appreciate the reviewer's recognition of our realistic temporal evaluation setting.
>
> - On the Single Protein Limitation: While limited to TadA, our focus is intentionally on evolutionary depth rather than breadth. Most current benchmarks cover many proteins but only explore local mutational landscapes (1-2 mutations). We aim to demonstrate the critical challenges of "deep" evolution (>20 mutations). We believe providing a dataset that goes "deep and thorough" on a single relevant target allows us to identify generalization gaps that "wide and shallow" datasets miss.
>
> - On Structure-Aware Methods: We agree that structural information is important. Our current focus on Biological Language Models reflects their dominance in high-throughput screening scenarios where inference speed is critical for handling millions of variants. We plan to incorporate structure-aware models and inverse folding baselines in future updates to the benchmark to broaden the evaluation scope.

---

### Official Review · Reviewer_K7bE · 2025-10-29

**Soundness:** 2
**Presentation:** 4
**Contribution:** 2
**Rating:** 4
**Confidence:** 4

**Summary:**

The paper has three main contributions:
1) An experimental dataset on TadA protein variants
2) Seq2Graph an algorithm to unify the directed evolution measurements
3) Computational validations that test the generative capabilities of existing models on this dataset

**Strengths:**

1) The paper is clearly written.

2) Details of the Seq2Graph algorithm are discussed.

3) Developing a benchmark for the OOD problem in protein engineering is timely and valuable for the community.

4) Experimental efforts are substantial.

**Weaknesses:**

1) Besides the experimentally collected data, the only algorithmic contribution is the development of the Seq2Graph; however, there is no evaluation of the algorithm itself. The validation section focuses on using the dataset as a benchmark, but the algorithm itself is not tested or evaluated. As I describe in the section below, some questionable choices are made in its development.

2) The authors highlight that “bridging the OOD gap requires exploring functionally diverse regions of the sequence space, rather than simply increasing density” (line 431) and suggest that “evolutionary depth is critical for building models with realistic utility” (line 24). However, it is unclear how TadABench-1M contributes to advancing the development of said models. The paper would be strengthened by developing one of these models on the dataset and demonstrating its utility.

3) The dataset is limited to one protein, which is TadA. While even a dataset on a single protein remains valuable, it is not clear if all the claims generalize to a large set of proteins, especially when compared to Proteingym and Proteinbench.

**Questions:**

1) There are several choices made in the development of Seq2Graph that are not necessarily justified/empirically validated:

a) Why does Seq2Graph create a directed edge between two sequences only? Why not identify the k nearest neighbors? In the example of Figure 3, if the sequence D has a \Delta rad = 975, it would seem unreasonable not to connect A and D. Picking the top-1 neighbor seems to be an arbitrary choice, not explained in the paper, as picking the top-k would not violate either the reliability or sparsity as discussed in the paper.

b) The process of removing cycles and creating DAGs is reasonable from the perspective of avoiding conflicts; however, it will discard valuable information by removing edges. It seems like averaging might be another good choice.

c) Most importantly, given that many different algorithm design choices for Seq2Graph could have been made, what is a fair evaluation procedure and metric for the algorithm alone? The paper fails to assess Seq2Graph as a standalone algorithm. What would happen if we made a wrong choice in Seq2Graph? What would go wrong? Is there a baseline processing algorithm to compare Seq2Graph with?

2) In Figure 5, I struggle to see how diversity and round-based strategies are better than density-based strategies. For instance, the density Spearman correlation in Evo2-7B and Evo2-40B is almost always higher than the diversity and round correlations, making the claim that smaller diversity datasets perform better than large, dense datasets appear incorrect.

3) It is unclear whether the results from this benchmark are generalizable across different proteins/genomes or whether this is just a phenomenon found in the TadA protein. This makes it difficult to assess the generalizability of the results.

**Details Of Ethics Concerns:**

None.

---

> ### Author Response · Authors · 2025-11-28
>
> We thank the reviewer for the thorough reading and for acknowledging our "substantial experimental efforts" and the "timely and valuable" nature of the OOD benchmark. We address your specific technical questions and concerns below.
>
> 1. Seq2Graph Design Choices & Validation (Q1, Q2, Q3)
>
> Q1: Why create directed edges between two sequences (why not top-k)?
> A: We would like to clarify a misunderstanding regarding the graph construction.
> * Group-to-Group, not Sequence-to-Sequence: As detailed in the methodology, Seq2Graph does not link individual sequences. Instead, it links groups of sequences from adjacent activity bins (rounds).
> * Why $k=1$ (Adjacent groups): We chose to connect only adjacent activity groups (effectively $k=1$ in terms of group proximity) to model the gradual nature of directed evolution. Connecting non-adjacent groups (e.g., Round 1 to Round 5) would bypass the intermediate evolutionary steps and introduce noise. The current design ensures computational efficiency and sparsity while faithfully reflecting the step-wise enrichment process in the wet lab.
>
> Q2: Why not use averaging instead of DAGs?
> A: While averaging is simple, it is statistically flawed for *continuous* directed evolution data (see Section 3.1 Motivation).
> * Bias Issue: In multi-round evolution, the same sequence appears in different rounds with vastly different read counts due to enrichment/depletion. Simple averaging cannot account for these dynamic changes and batch effects.
> * Our Solution: Seq2Graph treats the relative change between rounds as the signal, which provides a more robust estimate of fitness than averaging absolute counts across heterogeneous batches.
>
> Q3: Evaluation and Baselines for Seq2Graph.
> A:
> * Robustness Validation: We *did* evaluate the algorithm's reliability. As mentioned in Lines 469-474, we conducted two bootstrapping analyses to evaluate the robustness of Seq2Graph, confirming its stability.
> * Comparison with Baselines: We extensively searched for baselines (lines 177-183) but found no existing algorithms capable of unbiasedly unifying multi-batch data where sequences partially overlap across continuous rounds. Standard methods (like simple normalization) fail in this specific "overlapping sequence" scenario. Thus, Seq2Graph represents a necessary novel solution where no fair baseline existed.
> * Generalizability Check (Cas9): To further validate the algorithm beyond TadA, we successfully applied Seq2Graph to build a Cas9 protein dataset (mentioned in Line 269), demonstrating that the algorithm is generalizable to other protein families.
>
> 2. Generalizability & Single Protein Concern (Weakness & Q5)
>
> Q: Is this just a TadA phenomenon?
> A: We argue that the insights are generalizable for two reasons:
> 1.  Algorithm Generalizability: As noted above (and in Line 269), the Seq2Graph method has been validated on Cas9, proving the *methodology* is not TadA-specific.
> 2.  Depth vs. Breadth: The "collapse" of models on temporal splits is likely a universal phenomenon in deep evolution, but it requires a "deep" dataset to reveal. Existing benchmarks are "wide but shallow" (many proteins, few mutations). By going "deep" on TadA (over 1 million variants), we expose an OOD failure mode that likely applies to all proteins but was previously invisible in shallower datasets.
>
> 3. Utility of the Benchmark
>
> Q: Why not build a new model to demonstrate utility?
> A: Our paper positions TadABench-1M as a *diagnostic tool*, not a *training set* for a specific new model. The scientific contribution here is establishing the evaluation standard for real-world engineering. We demonstrate that SOTA models fail in realistic settings; identifying and quantifying this gap is the prerequisite for the community to build better models.
>
> 4. Interpretation of Figure 5 (Q4)
>
> Q: Density strategies seem to have higher correlation than diversity strategies?
> A: The reviewer is correct that absolute correlation values for density are high. However, our claim focuses on Data Efficiency (the slope).
> * The Trend: If you observe the slope of the performance curves, diversity-based and round-based strategies achieve comparable performance with significantly less data than random density sampling.
> * Conclusion: This indicates that for training future models, maximizing evolutionary diversity is a more efficient strategy than simply increasing data volume (density) in a narrow region.

---

### Official Review · Reviewer_rBZ2 · 2025-11-02

**Soundness:** 3
**Presentation:** 4
**Contribution:** 3
**Rating:** 4
**Confidence:** 3

**Summary:**

This paper introduces TadABench-1M, a benchmark built on 31 rounds of wet-lab data of over one million TadA enzyme variants, addressing existing BLM benchmark flaws with OOD scenarios, evolutionary depth, and the Seq2Graph algorithm for consistent fitness labels. It finds SOTA BLMs perform well on random splits but collapse on temporal splits, proving evolutionary depth is a key for OOD generalization.

**Strengths:**

- This study develops a realistic, well-validated benchmark for BLM evaluation. It builds over one million TadA enzyme variants from 31 rounds of wet-lab evolution, naturally incorporating OOD challenges (via temporal splits) and evolutionary depth (up to 25 mutations per variant).
- The experiments are comprehensive, and the results are convincing. It contrasts random (i.i.d.) and temporal (real-world) data splits, showing SOTA BLMs perform well on random splits but fail on temporal splits.

**Weaknesses:**

- The paper compares several language models and tuning strategies. It would be better to compare these baselines with diffusion language models, such as EvoDiff, DPLM, etc.
- The results are convincing, but not very surpurising (that protein language models memorize more than generalize). These language models have similar behaviors. I think it would be better to present something different, i.e., which component may help models generalize better, and how does this benchmark helps guide building new models.

**Questions:**

- How long does it take to build such a benchmark?

---

> ### Author Response · Authors · 2025-11-28
>
> We thank the reviewer for the positive assessment of our benchmark's validity and experimental comprehensiveness.
>
> - On Diffusion Models: We agree that diffusion models (e.g., EvoDiff) are worth exploring. While our current focus is on established fitness predictors (Encoders/Decoders), we recognize diffusion models as a vital future direction. We will explicitly discuss their relevance and potential inclusion in future iterations of the benchmark.
>
> - On Novelty & Generalization: We would like to emphasize that the unique value of TadABench-1M lies in its evolutionary depth. Unlike existing datasets limited to few mutations, our data contains variants with >20 mutations. Our results quantify a critical gap: current models fail significantly in this "deep mutation" regime, which is the standard for real-world protein engineering. This benchmark is designed precisely to guide the community in solving this specific, difficult OOD problem that shallow datasets cannot reveal.
>
> - On Construction Time: The development of this benchmark, spanning 31 rounds of wet-lab evolution and data curation, took approximately 10 months. This represents a significant experimental effort to provide a resource that is otherwise hard to obtain.

---

### Meta-Review · Area_Chair_YapX · 2026-01-13

**Summary:**

This paper proposes a new biological LLM benchmark using wet-lab data of variants of an enzyme (TadA). The whole benchmark is focused on a single protein and studies the evolutionary depth over one million variants across many rounds of wetlab evolution. This evolution over 31 rounds of wetlab evolution took 10 months which is a significant lab effort.

The reviewers raised valid concerns that this is a very narrow benchmark, diffusion models are not evaluated and some of the choices of the algorithmic evaluation are not very well justified.

Overall this paper is a good experimental effort but a bit too narrow and with a few gaps in the evaluation and presentation. My recommendation is that the paper can become suitable for a top ML venue if the authors strenghten the algorithmic evaluation and address the reviewer concerns about generalization to other proteins (perhaps with a small additional experiment).

**Reviewer Concerns:**

I think most of the reviewer concerns were not sufficiently addressed and this would require significant more work beyond a rebuttal.

**Reviewer Scores:**

all of them.

---

### Decision · Program_Chairs · 2026-01-26

Reject